# C-Reactive Protein/Albumin Ratio vs. Prognostic Nutritional Index as the Best Predictor of Early Mortality in Hospitalized Older Patients, Regardless of Admitting Diagnosis

**DOI:** 10.3390/nu17172907

**Published:** 2025-09-08

**Authors:** Cristiano Capurso, Aurelio Lo Buglio, Francesco Bellanti, Gaetano Serviddio

**Affiliations:** Department of Medical and Surgical Sciences, University of Foggia, Viale Luigi Pinto 1, 71122 Foggia, Italy; aurelio.lobuglio@unifg.it (A.L.B.); francesco.bellanti@unifg.it (F.B.); gaetano.serviddio@unifg.it (G.S.)

**Keywords:** older adults, in-hospital mortality, inflammation, malnutrition, CRP/albumin ratio, Prognostic Nutritional Index

## Abstract

**Background:** Malnutrition and systemic inflammation are major determinants of poor outcomes in hospitalized older adults, such as length of hospital stay (LOS), mortality, and readmission risk. The C-reactive protein to albumin ratio (CRP/Alb) and the Prognostic Nutritional Index (PNI) are simple biomarkers reflecting inflammation and nutritional status. Additionally, the PNI offers a straightforward method to assess both the nutritional state and mortality risk in older patients. **Objective:** The objective of this study was to compare the predictive accuracy of the CRP/Alb ratio and PNI for early in-hospital mortality at 7 and 30 days after admission in older patients, independent of admitting diagnosis. **Methods:** We retrospectively analyzed 2776 patients aged 65 years and older, admitted to the Internal Medicine and Aging Unit of the “Policlinico Riuniti” University Hospital in Foggia, Italy, between 2019 and 2025. Laboratory data at admission included CRP, albumin, and total lymphocyte count (TLC). The CRP/Alb ratio and PNI were calculated. Prognostic performance for 7- and 30-day mortality for both the CRP/Alb ratio and PNI was assessed using ROC curves, Cox regression, Kaplan–Meier survival analyses, and positive predictive value (PPV) comparisons, stratified by rehospitalization status and length of stay (LOS). The likelihood-ratio test was also performed to compare the 7- and 30-day mortality PPVs of the CRP/Alb ratio and the PNI, both for all patients and for re-hospitalized patients. **Results:** In-hospital mortality occurred in 444 patients (16%). Deceased patients showed significantly higher CRP/Alb ratios and lower PNI values than survivors (*p* < 0.001). Both the CRP/Alb ratio and PNI independently predicted 7- and 30-day mortality. A CRP/Alb ratio > 8 strongly predicted very early mortality (HR 10.46 for 7-day death), whereas a PNI < 38 predicted both 7- and 30-day mortality (HR 8.84 and HR 3.54, respectively). Among non-rehospitalized patients, the PNI demonstrated superior predictive performance regardless of LOS (*p* < 0.001). Among rehospitalized patients, the PNI was a more accurate predictor for short LOS (<7 days), while the CRP/Alb ratio performed better for longer LOS (≥7 days). **Conclusions:** Both the CRP/Alb ratio and PNI are inexpensive, readily available biomarkers for early risk stratification in hospitalized older adults. The CRP/Alb ratio is particularly effective in detecting very early mortality risk, while the PNI offers refined prognostic value across selected subgroups. Integrating these markers at admission may support personalized geriatric care and timely interventions.

## 1. Introduction

Population aging is a global trend with significant implications for healthcare systems. According to the World Health Organization (WHO), by 2030, one in six people worldwide will be 60 years of age or older; this population is expected to double by 2050, reaching 2.1 billion people. Notably, the population over 80 years of age is expected to triple during the same period, surpassing 426 million [1]. What is new is that two-thirds of the world’s population aged 60 years and over will live in low- and middle-income countries. The phenomenon of population aging began over 40 years ago in high-income countries. Among high-income countries, this demographic shift is especially pronounced in Japan, where it is estimated that, by 2030, the over-sixty-five-years age group will be 31.6% of the population [2], and Italy, where, in 2025, the over-sixty-five-years age group will be 21.6% of the population [3].

Despite increasing longevity, aging is a heterogeneous process. While some individuals remain physically and cognitively robust into their 80s, others exhibit early declines in function, often developing frailty, which is a multidimensional syndrome characterized by decreased physiological reserves and increased vulnerability or rather reduced resilience to stressors. Frail older adults often face negative health consequences, such as falls, cognitive decline, loss of independence, and multiple and prolonged hospital admissions, resulting in increased healthcare costs [4,5,6,7,8].

A common feature of frail older people is the significant increase in healthcare costs due to both the care burden and the numerous and repeated hospitalizations [9,10,11].

Malnutrition, particularly undernutrition, is a key contributor to frailty and poor clinical outcomes in hospitalized patients. It encompasses inadequate intake of energy or protein, micronutrient deficiencies, and significant weight loss [12]. In hospital settings, malnutrition is prevalent at admission and often worsens during the hospital stay. Estimates suggest that 15–70% of hospitalized individuals show evidence of undernutrition, depending on patient characteristics and care settings. Contributing factors include chronic disease burden, reduced appetite, complex medication regimens, and inadequate nutritional monitoring or intervention [13,14,15,16,17,18,19,20].

A significant percentage of patients arriving at the hospital show signs of malnutrition upon admission, which, in some cases, worsens during their stay.

Older patients admitted to acute care units already present with chronic diseases upon admission, including cognitive decline, malnutrition, complex home therapies, and multiple medications [21].

Adding to the objective fragility of these patients is the need for hospital medical procedures, whether diagnostic or therapeutic, which are sometimes non-priority and have a negative impact on their nutrition during hospitalization. Insufficient monitoring of the patient’s nutritional status is also common, often due to a lack of knowledge of standardized nutritional protocols [22,23,24].

Among older inpatients, malnutrition at the time of admission is one of the most important negative predictors of the risk of an increased incidence of complications, such as hospital-acquired infections or the development or worsening of pressure ulcers. Poor nutritional status also contributes to increased length of hospital stay (LOS), frequent rehospitalizations, poor response to treatment for the primary disease, and increased in-hospital mortality [25,26,27,28].

Albumin (Alb) is synthesized by the liver; it serves not only as a nutritional marker but also as an acute-phase reactant [29]. Low albumin levels have been associated with increased morbidity, prolonged hospitalization, and higher mortality, especially in older populations [30,31,32,33,34,35,36,37].

The assessment of serum Alb levels as a surrogate marker in measuring nutritional status is an established practice, since Alb is indicative of both hepatic synthesis and plasma distribution and overall protein loss, decreasing in hypercatabolic states [38,39,40,41,42,43,44], particularly in older patients.

Total lymphocyte count (TLC) is also a useful indicator of nutritional status. It is a readily available laboratory parameter, suitable for all age groups. Low TLC values (<1500/mm^3^), along with low serum Alb values, are indicative of malnutrition. Low TLC values are also associated with immune suppression, resulting in adverse outcomes, namely morbidity and mortality in hospitalized patients [45,46,47].

The Prognostic Nutritional Index (PNI) [48,49,50], calculated from Alb and TLC values [10 × Alb (g/dL)] + [0.005 × TLC (cells/mm^3^)], has been widely studied as a predictor of mortality across various clinical conditions, mainly cancer and surgical settings [51,52,53,54,55]. The PNI is a simple and rapid calculation tool that can help classify the conditions of older patients admitted to the hospital, based on their nutritional status, both to assess their risk of mortality, where the PNI is assessed at the time of admission, and to assess their risk of rehospitalization in the following thirty days, where the PNI is assessed at the time of discharge, since rehospitalization, in turn, constitutes a risk factor for adverse outcomes, namely death, irreversible loss of autonomy, or institutionalization.

On the other hand, systemic inflammation plays a central role in catabolic states, contributing to hypoalbuminemia and functional decline. C-reactive protein (CRP) is an inexpensive, sensitive marker of inflammation. Measuring CRP upon hospital admission, in addition to being indicative of acute inflammation, can help identify patients at increased risk for adverse short- and long-term outcomes, including mortality. Elevated CRP levels are significantly associated with an increased risk of all-cause mortality [56,57,58,59].

When considered in combination with albumin levels, the CRP/Albumin ratio (CRP/Alb ratio) reflects both nutritional and inflammatory status and is a strong predictor of early mortality, particularly in older hospitalized patients [60,61,62,63,64].

The CRP/Alb ratio, together with measurement of anthropometric parameters as part of a thorough clinical assessment, is an easily obtainable indicator of reduced energy and protein intake and adverse clinical outcomes among hospitalized older adults, even among those who are not severely ill [65].

The CRP/Alb ratio is a significant independent predictor of clinical outcomes among hospitalized older patients, regardless of admission diagnosis. The predictive value of this easily computable biomarker is of particular utility in geriatric care, where both systemic inflammation and malnutrition often coexist, both of which contribute to adverse outcomes [37,60,61,62,63,64,66,67,68,69].

The PNI has also been shown to be a significant predictor of clinical outcomes among hospitalized older patients, especially among patients hospitalized for acute heart failure or acute myocardial infarction, undergoing coronary artery bypass grafting [70,71,72,73], after hip fracture [74], with chronic kidney disease [75], or with cancer, for example, prostate cancer or lymphoma [76,77].

Having established that both the PNI and CRP/Alb ratio are accessible and objective tools that can improve risk assessment and clinical decision-making in hospitalized older patients, our retrospective observational study aimed to investigate the prognostic significance of the PNI and CRP/Alb ratio as predictors of hospital mortality at 7 and 30 days after admission and to compare their predictive performance in a population of older patients hospitalized in an acute care unit.

## 2. Materials and Methods

### 2.1. Study Population

Our study was conducted and reported in line with the TRIPOD recommendations [78]. A complete TRIPOD checklist is available in the Appendix A).

We retrospectively examined 4425 patients who were admitted to the Acute Care Unit of Internal Medicine and Aging, Policlinico Riuniti di Foggia, from 1 January 2019 to 29 May 2025. Key eligibility criteria for inclusion were (i) age ≥ 65 years; (ii) availability of admission laboratory tests, including C-reactive protein (CRP), serum albumin, and total lymphocyte count; and (iii) complete outcome ascertainment at 7 and 30 days. Key exclusion criteria were (i) patients younger than 65 years at admission; (ii) patients who were receiving albumin infusions at admission, to avoid confounding albumin levels; (iii) patients who left the hospital against medical advice; (iv) patients who were transferred to another department or acute care facility; (v) patients who were discharged to long-term care institutions; and (vi) patients who lacked complete admission lab data. No imputation was applied. These criteria ensured the final sample included older patients with complete and evaluable clinical and laboratory profiles, specifically regarding CRP, albumin, and TLC levels. In summary, from the initial group of 4425 patients, 1039 patients under the age of 65 years or receiving albumin infusions at admission were excluded, leaving a cohort of 3386 patients. From this group, we excluded 60 patients who left the hospital against medical advice, 324 patients who were transferred to another ward or the ICU, and 126 patients who were discharged to long-term care facilities. This step left 2876 patients. The final exclusion step removed 100 patients who had missing laboratory data at the time of admission. After exclusions, 2776 subjects were included in the final analysis. The flowchart steps are summarized in Figure 1.

### 2.2. Data Collection

For each patient, data were collected from the electronic medical record regarding serum C-reactive protein (CRP), total lymphocyte count (TLC), and albumin levels at admission, i.e., from the first blood draw within 6 h of admission. Additional variables included demographics (age, sex), length of hospital stay (LOS), and hospital outcome, i.e., discharge to home or in-hospital mortality at 7 and 30 days (primary outcomes).

We computed the CRP/Alb ratio as CRP (mg/L) divided by Alb (g/dL), following prior literature, and the PNI as [10 × Alb (g/dL)] + [0.005 × TLC (cells/mm^3^)], as described above.

All participants received standard medical treatment based on clinical assessment, which could include modification or continuation of home therapies. Information on in-hospital nutritional interventions (e.g., oral supplements, enteral/parenteral nutrition) and comorbidity indices (e.g., Charlson Index) was not consistently available in the retrospective source data and was, therefore, not included in this study. Measurements of functional status and GLIM criteria were also unavailable.

Albumin values were not corrected for serum calcium because total calcium was not consistently obtained at admission; we, therefore, report analyses using uncorrected albumin. A sensitivity analysis using corrected albumin will be planned in future prospective research where calcium is systematically collected.

Any hospital admission occurring within 30 days of a previous discharge was considered a rehospitalization event. Causes of death were classified according to ICD-9 criteria after having been obtained from the discharge records.

### 2.3. Statistical Analysis

We first summarized patient characteristics overall and by outcome status using appropriate descriptive statistics, i.e., mean ± SD for continuous variables and counts and percentages for categorical variables.

We assessed the distribution of the data using the Kolmogorov–Smirnov test, which indicated a non-Gaussian distribution (*p* < 0.001). Consequently, we used the Mann–Whitney U test (corrected by the Monte Carlo exact test) to compare continuous variables; Spearman’s correlation was used for bivariate analysis. Categorical comparisons (e.g., male vs. female, deceased vs. non-deceased) were assessed via chi-square tests with Monte Carlo correction. Effect sizes were calculated using Cohen’s d for means and Cohen’s h for proportions. Phi coefficient was applied when analyzing categorical associations between sex and survival status.

To assess the prognostic performance of the CRP/Alb ratio and the PNI for 7-day and 30-day in-hospital mortality, we performed receiver operating characteristic (ROC) curve analysis and calculated the optimal cut-off values. Optimal cut-offs were determined using the Youden Index from the ROC curve analysis. We performed logistic regression models for 7- and 30-day mortality, with the CRP/Alb ratio and PNI as main predictors, after adjusting for sex. Results were presented as odds ratio (OR) or hazard ratio (HR) with 95% CIs as appropriate to the model used in the main text. We selected 7 and 30 days as the outcome assessment time points, as 7 days represents the critical early phase of hospitalization, where acute mortality risk is highest, and 30 days is widely adopted as a standard clinical outcome measure in acute care and is consistent with previous literature.

Kaplan–Meier curves were plotted to illustrate survival probability at 7 and 30 days. Kaplan–Meier survival curves were compared with the log-rank test. Sensitivity analyses were conducted by LOS strata (≥7 days vs. <7 days) and included both overall and rehospitalized patients.

We performed a post hoc power analysis to detect the observed effect sizes for both the CRP/Alb ratio and PNI, with α = 0.05, at 7 and 30 days.

We also assessed model calibration using calibration plots (observed vs. predicted risk by deciles of predicted probability) and reported the calibration slope and intercept. Calibration plots and Hosmer–Lemeshow goodness–of–fit statistics are provided in the Appendix A).

All analyses were carried out using IBM SPSS Statistics version 25 and STATA SE version 14.2. Statistical significance was defined as *p* < 0.05.

## 3. Results

### 3.1. General Characteristics of the Study Cohort

The final study sample included 2776 patients, with women slightly outnumbering men (52.7% vs. 47.3%, *p* = 0.003). On average, female participants were older than male participants (*p* < 0.001). We did not find any statistically significant differences between female and males for either length of stay (LOS) or serum albumin levels. Males showed significantly elevated CRP levels (*p* = 0.001) and a higher CRP/Alb ratio (*p* = 0.005) compared to females. Conversely, total lymphocyte counts (TLC) were significantly higher in women (*p* = 0.001). The mean PNI did not differ significantly by sex (*p* = 0.098). Detailed sex-stratified characteristics are summarized in Table 1.

### 3.2. Age Stratification

To account for heterogeneity within the older population, we categorized patients into the following groups: “young-old” (65–74 years), “old” (75–84 years), and “very old” (≥85 years) [79,80]. Men were more prevalent in the youngest subgroup (*p* < 0.001), while women were predominant in the oldest age category (*p* < 0.001). No significant sex difference was observed in the 75–84 years age group (*p* = 0.705). Further stratified analyses by age group are planned for future work. Detailed age-stratified characteristics are summarized in Table 2.

### 3.3. Correlation Analyses

We performed Spearman correlation analysis between CRP, Alb, CRP/Alb ratio, TLC, and PNI. As expected, CRP and albumin were negatively correlated (r = −0.58, *p* < 0.001), CRP and CRP/Alb ratio were strongly positively correlated (r = 0.99, *p* < 0.001), and albumin and CRP/Alb ratio were inversely correlated (r = −0.66, *p* < 0.001). In addition, Alb and TLC were positively correlated (r = 0.21, *p* < 0.001), Alb and PNI were strongly positively correlated (r = 0.88, *p* < 0.001), and TLC and PNI were positively correlated (r = 0.58, *p* < 0.001). We also report the correlation coefficients after correction for sex and age at admission: CRP and albumin were negatively correlated (r = −0.49, *p* < 0.001), CRP and CRP/Alb ratio were strongly positively correlated (r = 0.95, *p* < 0.001), and albumin and CRP/Alb ratio were inversely correlated (r = −0.61, *p* < 0.001). In addition, Alb and TLC were positively correlated (r = 0.49, *p* = 0.010), Alb and PNI were positively correlated (r = 0.23, *p* < 0.001), and TLC and PNI were positively correlated (r = 0.50, *p* < 0.001). All results are reported in Table 3a,b.

Adjusting for sex, CRP and the CRP/Alb ratio both showed a significant positive correlation with LOS (*p* < 0.001). Serum albumin levels were inversely associated with LOS (*p* < 0.001). No meaningful correlation emerged between LOS and either patient age (*p* = 0.434) or TLC (*p* = 0.829). The PNI displayed a weak but statistically significant inverse correlation with LOS (*p* = 0.011). Full results are reported in Table 4.

### 3.4. Mortality and Associated Variables

A total of 444 patients (16%) died during their hospital stay. Consistent with data from the Italian National Registry of Causes of Death, managed by the National Institute of Statistics (ISTAT) [81,82], severe sepsis was the most frequently encountered cause of death (49.8%); subsequently, we found respiratory failure (9.9%) and respiratory infections with complications (8.1%). The main causes of death, for which a full breakdown is provided in Table 5, were reported for descriptive purposes only; therefore, we did not perform any inferential statistical comparisons.

Patients who died were older on average than those who survived (*p* < 0.001). No significant difference in mortality rate was observed between sexes (*p* = 0.276) or in LOS (*p* = 0.367). Deceased individuals exhibited significantly lower albumin levels and higher CRP values, resulting in elevated CRP/Alb ratios (all *p* < 0.001). TLC and PNI were significantly lower among those who died (*p* < 0.001). Descriptive statistics by survival status are shown in Table 6.

### 3.5. Predictive Performance and Survival Analyses

ROC analysis demonstrated that both the PNI and the CRP/Alb ratio significantly predicted in-hospital mortality. The PNI had an area under the curve (AUC) of 0.769 (Figure 2a), while the CRP/Alb ratio had an AUC of 0.730 (Figure 2b). Among patients with a hospital stay under seven days, the PNI’s AUC improved to 0.871 (Figure 2c), with an optimal cut-off = 38, consistent with previous findings [82], and the CRP/Alb ratio reached 0.890, with an optimal cut-off = 38 (Figure 2d); both AUC values were statistically significant (*p* < 0.001).

The CRP/Alb ratio threshold ≥ 8 showed strong predictive capability for 7-day mortality, with a sensitivity of 89% and specificity of 66% (Youden Index: 0.546). Similarly, a PNI < 38 predicted 7-day mortality with 87% sensitivity and 69% specificity (Youden Index: 0.562).

Kaplan–Meier survival estimates, compared by the log-rank test after adjusting for sex and stratifying by the CRP/Alb ratio, showed that a CRP/Alb ratio higher than 8 was associated with a reduction in survival (*p* < 0.001), both at 30 days (Figure 3a) and within 7 days from hospitalization (Figure 3b).

Cox regression analysis by the Breslow method, after adjusting for sex and stratifying for LOS, confirmed that a CRP/Alb ratio > 8 independently predicted 30-day mortality (hazard ratio, HR = 3.07) and was particularly impactful within the first 7 days (HR = 10.46).

Similarly, Kaplan–Meier survival estimates showed that a PNI value lower than 38 was associated with a reduction in survival (*p* < 0.001), both at thirty days (Figure 4a) and within seven days from hospitalization (Figure 4b).

Similarly, in Cox regression, a PNI < 38 was associated with an increased risk of death at 30 days (HR = 3.54) and, to an even greater extent, within 7 days (HR = 8.84).

### 3.6. Model Comparisons and Calibration

The calibration plots and Hosmer–Lemeshow goodness-of-fit test for the models, as shown in Appendix A, all demonstrate excellent performance. With predictive powers exceeding 90% and non-significant *p*-values from the Hosmer–Lemeshow test (*p* > 0.05), these models show a strong ability to discriminate between outcomes and are well-calibrated, indicating a good alignment between predicted probabilities and observed outcomes. In contrast, the model in Appendix A exhibits a low predictive power of 26%, which is visually confirmed by its calibration plot showing a significant deviation from the ideal line. This suggests a poor ability to accurately predict outcomes. While the corresponding Hosmer–Lemeshow test for Appendix A also shows a non-significant *p*-value (*p* = 0.3217), which suggests a good fit, this presents a notable contradiction between the low predictive power and the calibration test result.

Based on the calibration plots and Hosmer–Lemeshow goodness-of-fit test, a clear difference emerges in the predictive power of the CRP/Albumin ratio and the PNI. The CRP/Albumin ratio consistently demonstrates a high predictive power for mortality, exceeding 90% at both 7 and 30 days. This indicates its strong and reliable ability to discriminate between patient outcomes.

In contrast, the predictive power of the PNI is notably low at 7 days, at just 26%. However, its performance dramatically improves by day 30, where it also exceeds 90%. This suggests that the PNI’s predictive utility for mortality becomes much more significant over time.

For both markers, the Hosmer–Lemeshow goodness-of-fit tests show non-significant *p*-values, indicating that all models are well-calibrated, i.e., there is no significant difference between predicted probabilities and observed outcomes. While this is an expected finding for the models with high predictive power (CRP/Alb ratio at 7 and 30 days and PNI at 30 days), it presents a notable paradox for the PNI model at 7 days, which is well-calibrated despite its poor discriminatory power. This discrepancy may warrant further investigation.

### 3.7. Rehospitalized vs. Non-Rehospitalized Patients

Among the 369 rehospitalized patients (readmitted within 30 days of discharge), a CRP/Alb ratio > 8 was significantly associated with higher mortality risk, especially in early hospitalization (LOS < 7 days: odds ratio, OR = 16.03; LOS ≥ 7 days: OR = 6.69). Similarly, a PNI <3 8 yielded ORs of 48.4 and 3.93 for < 7 days and ≥ 7 days hospitalizations, respectively. These associations are outlined in Table 7.

A similar trend was found among non-rehospitalized patients (Table 8). A CRP/Alb ratio > 8 increased mortality risk, with OR = 15.02 for <7 days stays and OR = 2.75 for ≥7 days stays. A PNI < 38 also remained predictive (OR = 11.07 for <7 days stays; OR = 4.53 for ≥7 days stays).

We then compared the positive predictive value (PPV) of the CRP/Alb ratio and PNI using the likelihood-ratio test in both patient subgroups (rehospitalized and not). Among non-rehospitalized individuals (Table 9), the PNI outperformed the CRP/Alb ratio in predicting mortality for both LOS < 7 and ≥7 days (*p* < 0.001).

Among rehospitalized patients (Table 10), the PNI outperformed the CRP/Alb ratio in predicting mortality only among patients with LOS < 7 days (*p* = 0.002); among patients with LOS ≥ 7 days, the CRP/Alb ratio outperformed the PNI in predicting mortality (*p* < 0.001).

## 4. Discussion

The growing proportion of older adults poses substantial challenges to modern healthcare systems, primarily due to the increased prevalence of multimorbidity, frailty, and repeated hospitalizations. Undernutrition is a critical but often underrecognized contributor to negative health outcomes in this population. It frequently manifests through diminished protein and energy intake, deficiencies in essential nutrients, and progressive weight loss, all of which worsen during hospitalization. These nutritional deficiencies are strongly associated with extended hospital stays, recurrent admissions, treatment failure, and a heightened risk of in-hospital mortality.

### 4.1. Integration with Previous Literature

Our findings corroborate the role of the CRP/Alb ratio as a predictive marker of mortality across acute conditions, such as acute stroke and traumatic brain injury [83,84,85,86], and extend our previous research on the mortality predictive value of both the PNI and the CRP/Alb ratio [87,88].

In this large retrospective cohort study of 2776 hospitalized older adults, we assessed the prognostic accuracy of two commonly available indices—the PNI and the CRP/Alb ratio—in predicting short-term mortality at 7 and 30 days after hospital admission. We found that both the CRP/Alb ratio and the PNI were significant predictors of short-term in-hospital mortality.

Importantly, the CRP/Alb ratio emerged as the most robust predictor within the first 7 days of hospitalization. In contrast, the PNI retained superior prognostic value in selected subgroups, particularly in non-rehospitalized patients and in those with shorter lengths of stay. These findings highlight the complementary prognostic value of markers of systemic inflammation and nutritional status.

Our findings are consistent with prior studies showing that hypoalbuminemia and elevated CRP levels are independent predictors of adverse outcomes in older patients. The CRP/Alb ratio has been increasingly recognized as a simple but powerful biomarker integrating inflammation and nutritional reserve. Recent cohort studies [89,90] confirm that an elevated CRP/Alb ratio is associated with increased short-term mortality across diverse hospitalized populations. Likewise, the PNI has been validated as a reliable marker of nutritional and immunological status, with strong associations with mortality in older adults [91,92,93].

Our findings, which extend these observations by directly comparing the CRP/Alb ratio and PNI in a large, unselected cohort of older inpatients, indicate a clear relationship between altered biochemical markers and increased mortality risk. Patients who died during their hospitalization were older and exhibited significantly lower albumin levels and higher CRP values compared to survivors. The predominant cause of death was severe sepsis, aligning with prior literature on mortality drivers in older inpatients.

We also provide evidence that the CRP/Alb ratio captures the acute inflammatory burden that is particularly relevant for very early (≤7 days) mortality, whereas the PNI, which incorporates lymphocyte count, may be more reflective of baseline nutritional reserves and immune competence.

Albumin and total lymphocyte count (TLC) are widely recognized markers of nutritional and immune status, respectively. The PNI, derived from these two parameters, provides a useful snapshot of a patient’s nutritional reserve and immunocompetence, both of which are critical for survival in acute illness. Conversely, CRP reflects the systemic inflammatory response. The CRP/Alb ratio, by integrating nutritional and inflammatory information, offers a more holistic view of a patient’s vulnerability to poor outcomes.

In our cohort, a CRP/Alb ratio ≥ 8 significantly increased the likelihood of death within 7 days, while a PNI below 38 was also strongly predictive of early mortality. These results reinforce the clinical utility of both indices in identifying high-risk individuals shortly after admission.

A particularly novel aspect of our study was the comparative evaluation of these markers across distinct patient subgroups. Among non-rehospitalized individuals, the PNI outperformed the CRP/Alb ratio regardless of the duration of hospitalization. However, in the rehospitalized cohort, the CRP/Alb ratio showed greater predictive value for those with longer lengths of stay (≥7 days), whereas the PNI remained superior among patients with shorter hospitalizations. These differences highlight the importance of context when interpreting these indices and suggest a complementary role in clinical decision-making.

### 4.2. Clinical Implications

From a clinical perspective, considering that both the CRP/Alb ratio and PNI can be easily calculated from routine laboratory data at admission without additional costs or procedures, our findings further contribute to the already robust body of evidence suggesting that simple, routine lab-based indicators can support the early clinical risk assessment in older patients.

By integrating these markers into admission protocols, clinicians may be better equipped to early identify, particularly among older adults, frail patients at elevated risk and tailor interventions accordingly—particularly those aimed at improving nutritional support and mitigating inflammation.

Notably, a CRP/Alb ratio ≥ 8 and PNI < 38 identified subgroups at particularly high risk. Such thresholds could be adopted in clinical practice to trigger alerts within electronic health records, prompting clinicians to evaluate patient care strategies more intensively during the critical early days of hospitalization.

### 4.3. Strengths

The strengths of this study include the large sample size (n = 2776); the focus on an unselected, real-world cohort of hospitalized older adults; and the direct head-to-head comparison of two established prognostic indices. This study also provides subgroup analyses stratified by length of stay and by rehospitalization and additional analyses, including a calibration assessment.

### 4.4. Limitations

Several limitations must be acknowledged: (i) the retrospective nature of the analysis may introduce information bias and limit causal inference, and being a single-center study conducted in Southern Italy, generalizability to other healthcare settings and populations may be limited; (ii) standardized comorbidity indices (e.g., Charlson Comorbidity Index) and illness severity scores (e.g., APACHE II, SOFA) were not systematically available in the dataset, and as a result, residual confounding by underlying comorbid conditions or acute illness severity cannot be excluded; (iii) data on body mass index (BMI), muscle mass, GLIM-defined malnutrition, and functional status (activities of daily living, mobility) were not collected, and therefore, we could not directly compare the CRP/Alb ratio or PNI with comprehensive geriatric or nutritional assessments; (iv) albumin values were not corrected for serum calcium, as calcium was inconsistently available at admission, and this may have led to minor misclassification of albumin-related indices; (v) information on in-hospital nutritional support (oral supplementation, enteral/parenteral nutrition), anti-inflammatory treatments, or other therapeutic interventions was unavailable, limiting interpretation of how management strategies may have influenced outcomes; (vi) this study was restricted to in-hospital mortality at 7 and 30 days, and longer-term outcomes, such as 90-day mortality, functional decline, or subsequent rehospitalizations, were not assessed; (vii) patients with missing laboratory values at admission were excluded using listwise deletion, and although this approach maintains internal consistency, it may introduce selection bias if excluded patients differed systematically from included patients.

Despite these limitations, this study included robust statistical analyses, which strengthen the reliability of the main findings, due to the large sample size and systematically collected laboratory data at admission.

### 4.5. Future Directions

Future research should seek to address these gaps by incorporating more comprehensive clinical assessments and tracking longitudinal biomarker trends.

Notably, prospective, multicenter studies are needed to confirm the prognostic role of the CRP/Alb ratio and PNI in hospitalized older adults. Future work should incorporate standardized comorbidity indices, nutritional assessments (GLIM), and functional measures as well as evaluate the potential impact of early nutritional and anti-inflammatory interventions on modifying these biomarkers and improving outcomes.

## 5. Conclusions

This large prospective study provides us with further confirmation regarding the predictive relevance of both the PNI and CRP/Alb ratio in the evaluation of early hospital mortality among older patients.

Although each marker individually contributes valuable information, its utility varies depending on clinical circumstances.

Notably, both the CRP/albumin ratio (CRP/Alb) and the Prognostic Nutritional Index (PNI) were significant, easily obtainable predictors of early in-hospital mortality. The CRP/Alb ratio was particularly effective for identifying patients at risk of very early mortality (≤7 days), while the PNI retained greater value in selected subgroups and over the 30-day horizon. These findings suggest that the CRP/Alb ratio and PNI capture complementary aspects of the interplay between inflammation and malnutrition in acute care geriatric medicine.

Our results suggest that these accessible and low-cost biomarkers could play a central role in hospital triage and geriatric care planning.

Namely, because both indices are derived from routine laboratory tests available at admission, they represent cost-effective and practical tools for early risk stratification. Identifying older adults at high risk for short-term mortality may allow clinicians to implement closer monitoring, individualized care planning, and timely nutritional or anti-inflammatory interventions during the critical early phase of hospitalization.

Despite the limitations related to study design and data availability, this analysis supports their integration into early risk stratification protocols.

Future prospective multicenter studies incorporating standardized comorbidity indices, GLIM-defined malnutrition, and functional assessments are warranted to validate these results and to explore whether targeted interventions can improve outcomes among high-risk patients identified by the CRP/Alb ratio or PNI.

## Figures and Tables

**Figure 1 nutrients-17-02907-f001:**
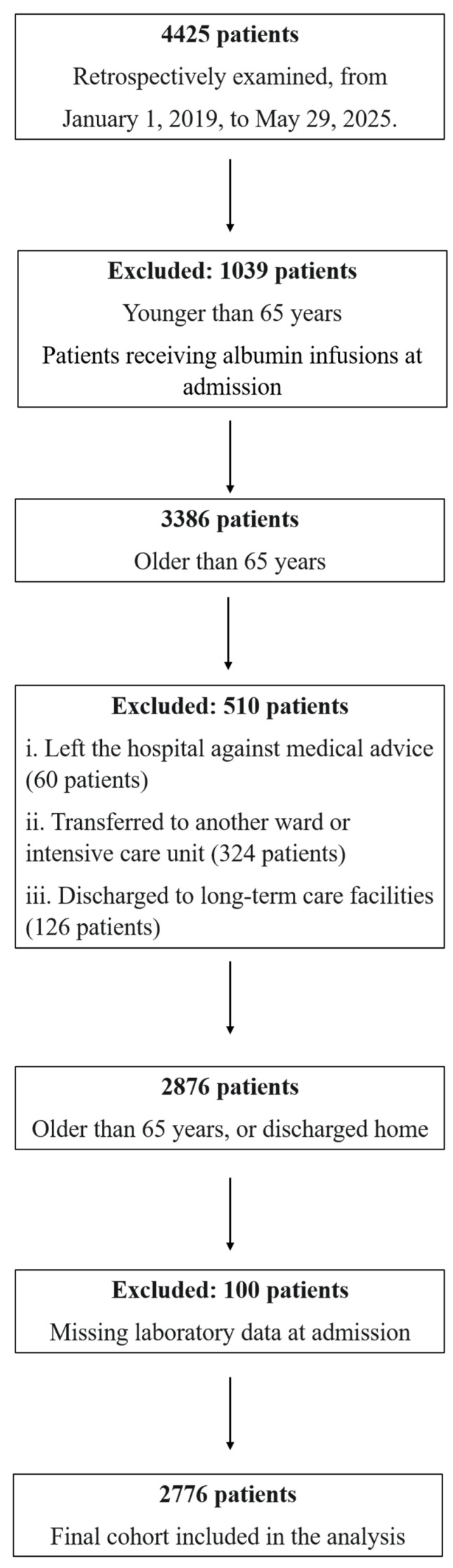
Flowchart steps from the starting population to the final cohort included in the analysis.

**Figure 2 nutrients-17-02907-f002:**
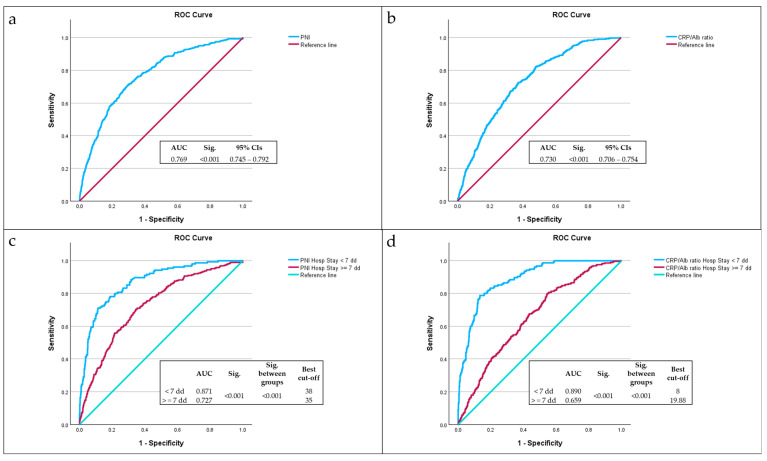
Analysis of ROC curves of PNI (**a**), CRP/Alb ratio (**b**), CRP/Alb ratio among LOS < 7 or ≥7 days (**c**), and CRP/Alb ratio among LOS < 7 or ≥7 days (**d**) as predictors of mortality, compared with the reference line (red line).

**Figure 3 nutrients-17-02907-f003:**
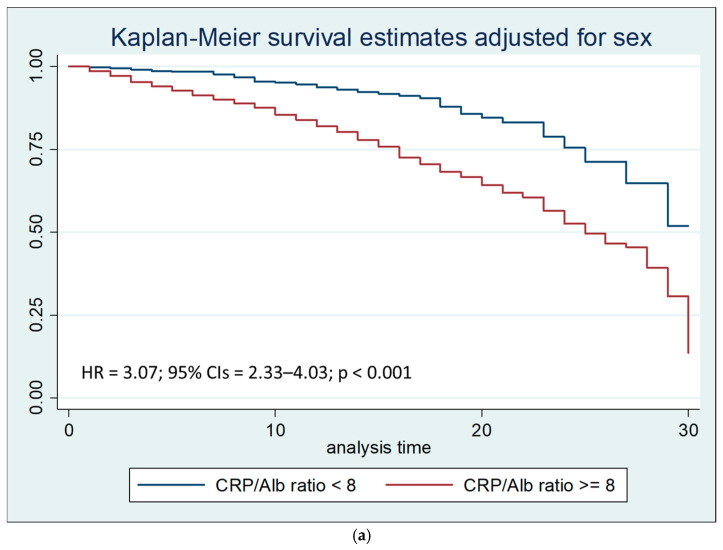
(**a**) Kaplan–Meier survival estimates stratified by CRP/Alb ratio < 8 vs. CRP/Alb ratio ≥ 8 after 30 days of hospitalization (chi-square 82.26; *p* < 0.001). (**b**) Kaplan–Meier survival estimates stratified by CRP/Alb ratio < 8 vs. CRP/Alb ratio ≥ 8 within 7 days of hospitalization (chi-square 140.29; *p* < 0.001).

**Figure 4 nutrients-17-02907-f004:**
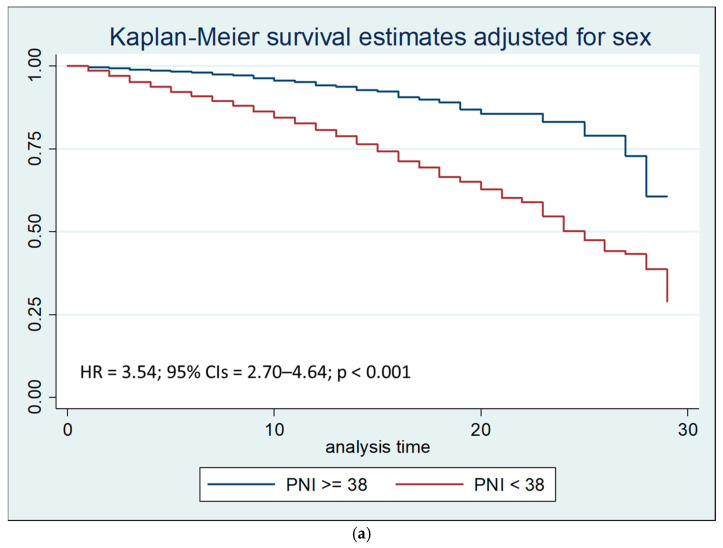
(**a**) Kaplan–Meier survival estimates stratified by PNI < 38 vs. PNI ≥ 38 after 30 days of hospitalization (chi-square 108.82; *p* < 0.001). (**b**) Kaplan–Meier survival estimates stratified by PNI < 38 vs. PNI ≥ 38 within 7 days of hospitalization (chi-square 138.31; *p* < 0.001).

**Table 1 nutrients-17-02907-t001:** General characteristics of the cohort, after stratification by sex, indicated as number (%) or as mean ± SD.

	Males	Females	Effect Size Cohen’s h	Sig. (99% CIs)
**Subjects N (%)**	1314 (47.3%)	1462 (52.7%)	0.11	0.003 (0.000–0.006)
**Age at hospitalization** **(Mean ± SD)**	79 ± 8	82 ± 8	0.38	<0.001 (0.000–0.002)
**LOS** **(Mean ± SD)**	11 ± 7	11 ± 7	0.00	0.489 (0.464–0.513)
**Albuminemia** **(Mean ± SD)**	3.05 ± 0.6	3.03 ± 0.6	0.03	0.329 (0.306–0.352)
**CRP** **(Mean ± SD)**	80.2 ± 89	73.3 ± 88	0.08	0.001 (0.000–0.003)
**TLC** **(Mean ± SD)**	1405.3 ± 1886.5	1542.2 ± 3598.4	0.05	0.001 (0.000–0.003)
**CRP/Alb ratio** **(Mean ± SD)**	31.3 ± 38	29.3 ± 40	0.05	0.005 (0.001–0.008)
** PNI ** **(Mean ± SD)**	37.6 ± 12	39.0 ± 39	0.05	0.098 (0.084–0.113)

**Table 2 nutrients-17-02907-t002:** Features of the cohort, after being stratified by age subgroups.

	Males	Females	Effect Size Cohen’s h	Sig. (99% CIs)
**Young**-**old N (%)** **(65–74 years)**	381 (57%)	288 (43%)	0.28	<0.001 (0.000–0.007)
**Old N (%)** **(75–84 years)**	551 (49.4%)	564 (50.6%)	0.02	0.705 (0.670–0.740)
**Very Old N (%)** **(85+ years)**	382 (38.5%)	610 (61.5%)	0.46	<0.001 (0.000–0.005)

**Table 3 nutrients-17-02907-t003:** (**a**) Spearman’s correlation among CRP, serum albumin, CRP/Alb ratio, TLC, and PNI. (**b**) Partial correlation, corrected by sex and age at admission, among CRP, serum albumin, CRP/Alb ratio, TLC, and PNI.

(a)						
		**CRP**	**Albuminemia**	**CRP/Alb Ratio**	**TLC**	**PNI**
**CRP**	Spearman’s correlation	1	−0.577	0.992	−0.284	−0.584
	Significance (2-tailed)		<0.001	<0.001	<0.001	<0.001
**Albuminemia**	Spearman’s correlation	−0.577	1	−0.662	0.21	0.888
	Significance (2-tailed)	<0.001		<0.001	<0.001	<0.001
**CRP/Alb ratio**	Spearman’s correlation	0.992	−0.662	1	−0.286	−0.655
	Significance (2-tailed)	<0.001	<0.001		<0.001	<0.001
**TLC**	Spearman’s correlation	−0.284	0.21	−0.286	1	0.587
	Significance (2-tailed)	<0.001	<0.001	<0.001		<0.001
**PNI**	Spearman’s correlation	−0.584	0.888	−0.655	0.587	1
	Significance (2-tailed)	<0.001	<0.001	<0.001	<0.001	
**(b)**						
		**CRP**	**Albuminemia**	**CRP/Alb ratio**	**TLC**	**PNI**
**CRP**	Partial correlation	1	−0.489	0.949	−0.054	−0.143
	Significance (2-tailed)		<0.001	<0.001	<0.001	<0.001
**Albuminemia**	Partial correlation	−0.489	1	−0.610	0.049	0.232
	Significance (2-tailed)	<0.001		<0.001	0.01	<0.001
**CRP/Alb ratio**	Partial correlation	0.949	−0.610	1	−0.052	−0.168
	Significance (2-tailed)	<0.001	<0.001		<0.001	<0.001
**TLC**	Partial correlation	−0.054	0.049	−0.052	1	0.587
	Significance (2-tailed)	<0.001	0.01	<0.001		<0.001
**PNI**	Partial correlation	−0.143	0.232	−0.168	0.506	1
	Significance (2-tailed)	<0.001	<0.001	<0.001	<0.001	

**Table 4 nutrients-17-02907-t004:** Partial correlation, corrected by sex, between LOS and age at admission, CRP, serum albumin, TLC, CRP/Alb ratio, and PNI.

		Length of Stay
**Age at admission**	Partial correlation	−0.015
	Significance (2-tailed)	0.434
** CRP **	Partial correlation	0.148
	Significance (2-tailed)	<0.001
** Albuminemia **	Partial correlation	−0.189
	Significance (2-tailed)	<0.001
** TLC **	Partial correlation	−0.004
	Significance (2-tailed)	0.829
**CRP/Alb ratio**	Partial correlation	0.150
	Significance (2-tailed)	<0.001
** PNI **	Partial correlation	−0.048
	Significance (2-tailed)	0.011

**Table 5 nutrients-17-02907-t005:** Main causes of death, expressed as number (%), among study patients.

	Deceased
Severe sepsis N (%)	221 (49.8%)
Pulmonary edema and respiratory failure N (%)	44 (9.9%)
Any respiratory infection and inflammation with complications N (%)	36 (8.1%)
Pleural effusion with complications N (%)	19 (4.3%)
Heart failure and shock N (%)	19 (4.3%)
Malignant neoplasms of digestive system with complications N (%)	9 (2.0%)
Any infectious disease N (%)	9 (2.0%)
Severe renal failure N (%)	8 (1.8%)
All other causes N (%)	79 (17.8%)
**Total number of deaths N (%)**	444 (100.0%)

**Table 6 nutrients-17-02907-t006:** Clinical features of patients, expressed as number (%) or as mean ± SD, after being stratified by deceased and not deceased.

	Deceased	Non-Deceased	Effect Size Cohen’s h	Sig. (99% CIs)
**Subjects N (%)**	444 (16%)	2332 (84%)	1.50	<0.001 (0.000–0.002)
	** Deceased **	** Non-Deceased **	**Effect size** **Phi Coefficient**	** Sig. **
**Male N (%)**	221 (49.8%)	1093 (46.9%)	0.02	0.276
**Female N (%)**	223 (50.2%)	1239 (53.1%)		
	** Deceased **	** Non-Deceased **	**Effect size** **Cohen’s d**	** Sig. ** **(99% CIs)**
**Age at hospitalization** **(Mean ± SD)**	84 ± 8	80 ± 8	0.5	<0.001 (0.000–0.002)
**LOS** **(Mean ± SD)**	12 ± 10	11 ± 7	0.12	0.367 (0.344–0.391)
**Albuminemia** **(Mean ± SD)**	2.6 ± 0.6	3.1 ± 0.6	0.8	<0.001 (0.000–0.002)
**CRP** **(Mean ± SD)**	123.6 ± 95.9	67.6 ± 84.2	0.62	<0.001 (0.000–0.002)
** TLC ** **(Mean ± SD)**	1432.7 ± 6016	1485.9 ± 1802	0.01	<0.001 (0.000–0.002)
** PNI ** **(Mean ± SD)**	32.7 ± 30.6	39.4 ± 29.5	0.22	<0.001 (0.000–0.002)
**CRP/Alb ratio** **(Mean ± SD)**	54.8 ± 47.8	25.6 ± 35.8	0.69	<0.001 (0.000–0.002)

**Table 7 nutrients-17-02907-t007:** CRP/Alb ratio ≥ 8, PNI < 38, LOS, and risk of death among re-hospitalized patients.

CRP/Alb Ratio ≥ 8	OR	95% CIs	Sig.
**LOS ≥ 7 days**	6.69	2.53–22.22	<0.001
**LOS < 7 days**	16.03	4.29–87.16	
**PNI < 38**	**OR**	**95% CIs**	**Sig.**
**LOS ≥ 7 days**	3.93	1.66–10.75	<0.001
**LOS < 7 days**	48.4	7.08–2012.05	

**Table 9 nutrients-17-02907-t009:** Best PPV between CRP/Alb ratio and PNI among non-rehospitalized patients.

	PPV			
	CRP/Alb Ratio (%)	NPI (%)	LR chi2	Sig.
**All patients**	20.35%	22.81%	31.2	<0.001
**LOS ≥ 7 days**	16.13%	18.57%	6.9	0.009
**LOS < 7 days**	37.94%	39.76%	43.44	<0.001

**Table 10 nutrients-17-02907-t010:** Best PPV between CRP/Alb ratio and PNI among rehospitalized patients.

	PPV REOSP			
	CRP/Alb Ratio (%)	NPI (%)	LR chi2	Sig.
**All patients**	37.50%	36.33%	21.57	<0.001
**LOS ≥ 7 days**	32.98%	31.09%	13.6	<0.001
**LOS < 7 days**	51.67%	52.38%	9.25	0.002

**Table 8 nutrients-17-02907-t008:** CRP/Alb ratio ≥ 8, PNI < 38, LOS, and risk of death among non-rehospitalized patients.

CRP/Alb Ratio ≥ 8	OR	95% CIs	Sig.
**LOS ≥ 7 days**	2.75	1.92–3.98	<0.001
**LOS < 7 days**	15.02	8.24–29.14	
**PNI < 38**	**OR**	**95% CIs**	**Sig.**
**LOS ≥ 7 days**	4.53	3.11–6.72	<0.001
**LOS < 7 days**	11.07	6.60–19.11	

## Data Availability

Data is unavailable due to privacy restrictions.

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
