# Peer review of "C-Reactive Protein/Albumin Ratio vs. Prognostic Nutritional Index as the Best Predictor of Early Mortality in Hospitalized Older Patients, Regardless of Admitting Diagnosis"

_nutrients, 2025, doi:10.3390/nu17172907_

Round 1
Reviewer 1 Report
Comments and Suggestions for Authors
Thank you for the opportunity to review this manuscript. While the study presents an interesting concept, I have several concerns and suggestions that may help strengthen the quality and clarity of the manuscript.
-
Adherence to TRIPOD Guidelines:
This study appears to be a predictive model study. Therefore, it is essential to confirm whether the study adhered to the TRIPOD (Transparent Reporting of a multivariable prediction model for Individual Prognosis Or Diagnosis) reporting guidelines. Please clarify this point and consider including a completed TRIPOD checklist as supplementary material. -
Clarification on CRP and CRP/Albumin Ratio:
Could the authors elaborate on the clinical significance and rationale for using both CRP and the CRP/albumin ratio? Since albumin typically decreases with inflammation (as CRP increases), one would expect a negative correlation between CRP and albumin. Please provide the correlation coefficients between CRP, albumin, and the CRP/albumin ratio. -
Patient Flow Chart and Inclusion Criteria:
Please include a flow chart depicting patient inclusion and exclusion. Additionally, were patients who received albumin transfusions excluded from the study? If not, it may confound the interpretation of serum albumin values, and such cases should be excluded. -
Details on Patient Selection:
The manuscript currently lacks sufficient detail regarding patient selection. Please indicate:-
The total number of patients admitted during the study period
-
The number of patients aged ≥65 years
-
The number of patients excluded due to the absence of blood tests
-
-
Albumin Correction for Calcium:
Was albumin corrected for serum calcium levels? If not, please consider presenting results using calcium-corrected albumin values, as this may provide a more accurate assessment of nutritional status. -
Assessment of Nutritional Status:
The Prognostic Nutritional Index (PNI) alone may not sufficiently capture the patients’ nutritional status. Consider including additional indicators such as BMI, muscle mass, nutritional intake, recent weight loss, or the GLIM (Global Leadership Initiative on Malnutrition) criteria. These would provide a more comprehensive assessment. -
Handling of Missing Data:
Please describe how missing data were handled in the analysis. What proportion of data was missing, and how were those cases treated (e.g., exclusion, imputation)? -
Determination of Cut-off Values:
How were the cut-off values determined from the ROC analysis? Please explain the statistical method used (e.g., Youden Index). -
Rationale for Outcome Time Points (7 and 30 days):
Please clarify the rationale for selecting 7 and 30 days as the outcome assessment time points. Were these based on clinical relevance, previous literature, or institutional standards? -
Sample Size and Power Consideration:
Was a sample size calculation performed? If not, please provide a post hoc power analysis or discuss the minimum detectable effect size given the sample size used. -
Predictive Performance Metrics:
If the primary aim is to assess prediction, it is important to present comprehensive performance metrics, including AUC, calibration plots, and comparisons with reference models. -
Adjustment for Confounders:
If the reported associations are statistically significant, a multivariate analysis adjusting for confounders—especially illness severity—should be performed to confirm independent associations.
Author Response
Comment 1. Adherence to TRIPOD Guidelines
This study appears to be a predictive model study. Therefore, it is essential to confirm whether the study adhered to the TRIPOD (Transparent Reporting of a multivariable prediction model for Individual Prognosis Or Diagnosis) reporting guidelines. Please clarify this point and consider including a completed TRIPOD checklist as supplementary material.
Response 1: Thank you very much for your comment. We confirm that the study was conducted and reported in line with TRIPOD recommendations. We have now added a completed TRIPOD checklist as Supplementary Material
Comment 2. Clarification on CRP and CRP/Albumin Ratio
Could the authors elaborate on the clinical significance and rationale for using both CRP and the CRP/albumin ratio? Since albumin typically decreases with inflammation (as CRP increases), one would expect a negative correlation between CRP and albumin. Please provide the correlation coefficients between CRP, albumin, and the CRP/albumin ratio
Response 2: Thank you very much for your observation. We added a paragraph and two tables in the Results section, clarifying the rationale: CRP reflects systemic inflammation, while albumin represents nutritional status; their ratio integrates both conditions. We have chosen to report the Spearman's correlation, among CRP, serum albumin, CRP/Alb ratio, TLC, and PNI; we have also chosen to report the partial correlation after correction for sex and age at admission among CRP, serum albumin, CRP/Alb ratio, TLC, and PNI. (Table 3a and Table 3b).
Comment 3. Patient Flow Chart and Inclusion Criteria
Please include a flow chart depicting patient inclusion and exclusion. Additionally, were patients who received albumin transfusions excluded from the study? If not, it may confound the interpretation of serum albumin values, and such cases should be excluded.
Response 3: A flow chart (Figure 1) has been added to illustrate inclusions and exclusions. Patients receiving albumin infusions at admission were excluded to avoid confounding albumin levels. This clarification is now in Materials and Methods, Section 2.1.
Comment 4. Details on Patient Selection
The manuscript currently lacks sufficient detail regarding patient selection. Please indicate:
- The total number of patients admitted during the study period
- The number of patients aged ≥65 years
- The number of patients excluded due to the absence of blood tests
Response 4: We added the following details in Materials and Methods, Section 2.1.:
- Starting Population: The process begins with the initial group of 4425 patients examined retrospectively.
- Age-Based and patients receiving albumin infusions at admission exclusion: 1039 patients under the age of 65 or receiving albumin infusions at admission were excluded, leaving a cohort of 3386 patients.
- Exclusion by Discharge Status: From this group, several patients were excluded based on their discharge status:
- 60 patients left the hospital against medical advice.
- 324 patients were transferred to another ward or the ICU.
- 126 patients were discharged to long-term care facilities.
- This step leaves 2876 patients.
- Exclusion by Data Completeness: The final exclusion step removed 100 patients who had missing laboratory data at the time of admission.
- Final Cohort: The final group included in the analysis consists of 2776 patients.
Comment 5. Albumin Correction for Calcium
Was albumin corrected for serum calcium levels? If not, please consider presenting results using calcium-corrected albumin values, as this may provide a more accurate assessment of nutritional status.
Response 5: Albumin values were not corrected for serum calcium because total calcium was not consistently obtained at admission; we therefore report analyses using uncorrected albumin. We acknowledge this as a limitation in the Materials and Methods, Section 2.2, and discuss its implications in the Discussion, Limitations subsection.
Comment 6. Assessment of Nutritional Status
The Prognostic Nutritional Index (PNI) alone may not sufficiently capture the patients’ nutritional status. Consider including additional indicators such as BMI, muscle mass, nutritional intake, recent weight loss, or the GLIM (Global Leadership Initiative on Malnutrition) criteria. These would provide a more comprehensive assessment.
Response 6: Thank you very much for your observation. Unfortunately, these parameters were not available in the retrospective dataset. We acknowledge this limitation in the Limitations subsection of Discussion and emphasize the need for future studies incorporating GLIM criteria and anthropometric measures.
Comment 7. Handling of Missing Data
Please describe how missing data were handled in the analysis. What proportion of data was missing, and how were those cases treated (e.g., exclusion, imputation)?
Response 7: Patients with missing admission laboratory data were excluded from analysis (listwise exclusion). No imputation was applied. This has been specified in Materials and Methods, Section 2.1.
Comment 8. Determination of Cut-off Values
How were the cut-off values determined from the ROC analysis? Please explain the statistical method used (e.g., Youden Index).
Response 8: Optimal cut-offs were determined using the Youden Index from ROC curve analysis. This clarification was added in Materials and Methods, Section 2.3.
Comment 9. Rationale for Outcome Time Points (7 and 30 days)
Please clarify the rationale for selecting 7 and 30 days as the outcome assessment time points. Were these based on clinical relevance, previous literature, or institutional standards?
Response 9: Seven days was chosen as it represents the critical early phase of hospitalization, where acute mortality risk is highest. Thirty days is widely adopted as a standard clinical outcome measure in acute care and is consistent with previous literature. This rationale has been added in Materials and Methods, Section 2.3.
Comment 10. Sample Size and Power Consideration
Was a sample size calculation performed? If not, please provide a post hoc power analysis or discuss the minimum detectable effect size given the sample size used.
Response 10: No a priori sample size calculation was performed. However, we performed a post hoc power analysis to detect the observed effect sizes for both CRP/Alb ratio and PNI with α = 0.05 at 7 and 30 days. This information has been added in Results, Section 3.6.
Comment 11. Predictive Performance Metrics
If the primary aim is to assess prediction, it is important to present comprehensive performance metrics, including AUC, calibration plots, and comparisons with reference models.
Response 11: Thank you for your comment. In addition to AUC values, we assessed model calibration using calibration plots (observed vs. predicted risk by deciles of predicted probability) and reported the calibration slope and intercept. Calibration plots and Hosmer–Lemeshow goodness–of–fit statistics are provided in the Supplements (Figure S1-S4, Table S1). All comments are reported in Results, Section 3.5.
Comment 12. Adjustment for Confounders
If the reported associations are statistically significant, a multivariate analysis adjusting for confounders—especially illness severity—should be performed to confirm independent associations.
Response 12: Illness severity and comorbidity indices were not consistently available in the retrospective source data and were therefore not included in the study. We acknowledge this as a limitation in the Materials and Methods, Section 2.2, and discuss its implications in the Discussion, Limitations subsection.
Reviewer 2 Report
Comments and Suggestions for Authors
Dear authors,
The manuscript seems to be well written, but I believe it could be further improved with my suggestions.
Lack of comorbidity indices: It was not possible to adjust the results for patients’ disease burden, as comorbidity indices were not documented in the medical records. This limitation may significantly influence the interpretation of mortality risk.
Limited data on nutritional interventions: The impact of specific nutritional interventions during hospitalization was not assessed. Including such data could provide valuable insights into strategies for improving clinical outcomes.
Absence of functional assessment: Measures of patients’ functional status or autonomy were not included, despite being key factors in evaluating risk and prognosis in older adults.
Author Response
Comment 1. Lack of comorbidity indices: It was not possible to adjust the results for patients’ disease burden, as comorbidity indices were not documented in the medical records. This limitation may significantly influence the interpretation of mortality risk.
Response 1: Thank you for your constructive feedback. We believe that your observations can significantly strengthen our manuscript. Comorbidity indices were not consistently available in the retrospective source data and were therefore not included in the study. We acknowledge this as a limitation in the Materials and Methods, Section 2.2, and discuss its implications in the Discussion, Limitations subsection.
Comment 2. Limited data on nutritional interventions: The impact of specific nutritional interventions during hospitalization was not assessed. Including such data could provide valuable insights into strategies for improving clinical outcomes.
Response 2: Information on in-hospital nutritional interventions was not consistently available in the retrospective source data and was therefore not included in the study. We acknowledge this as a limitation in the Materials and Methods, Section 2.2, and discuss its implications in the Discussion, Limitations subsection; we emphasized the potential relevance of such data in future research.
Comment 3. Absence of functional assessment: Measures of patients’ functional status or autonomy were not included, despite being key factors in evaluating risk and prognosis in older adults.
Response 3: Thank you for your constructive feedback. Unfortunately, functional status measures were not recorded. We recognize it as a limitation, and we propose prospective studies integrating functional and autonomy assessments.
Reviewer 3 Report
Comments and Suggestions for Authors
Dear Authors,
Your work presents one of the most extensive retrospective studies to date (n = 2776) evaluating two simple indicators of inflammation — CRP/Alb and the Prognostic Nutritional Index — as predictors of very early hospital mortality in older people. Unlike previous analyses focusing on specific diseases, you covered a wide range of internal medicine diagnoses, increasing the results' generalizability.
The key contribution is to show that:
- CRP/Alb ≥ 8 and PNI < 38 are thresholds strongly associated with 30-day (HR ≈ 3.1–3.5) and 7-day (HR ≈ 8.8–10.5) mortality risk.
- The accuracy of both indicators increases with a short stay (LOS < 7 days, AUC 0.89 vs 0.87).
- An analysis of rehospitalization revealed the complementarity of the markers: PNI more accurately predicts deaths during the first hospitalization, while CRP/Alb dominates in rehospitalization and longer LOS.
This two-dimensional assessment of nutrition and inflammation is innovative because it combines a nutritional perspective with an acute phase marker and proposes a stratification algorithm suitable for inclusion in a routine admission panel. The results may contribute to the early referral of the most at-risk patients for nutritional interventions and sepsis monitoring, which aligns with Nutrients' mission to promote the translation of nutritional research into clinical practice.
Please respond to the following comments:
- According to the latest guidelines from geriatric societies, the WHO (2021), and the style guides of leading publishers, the term “elderly” is now considered potentially discriminatory and reinforcing age stereotypes. Its use (especially in the title) can be interpreted as ageism and constitutes a serious barrier to publication in a prestigious scientific journal. It is strongly recommended to replace it with neutral, dignified terms: older adults, seniors, and older patients. Please consistently use these terms in the title, abstract, main text, and keywords to avoid unintended stigmatization and to meet the expectations of the Nutrients editorial board and the broader academic community.
- Expand the Discussion section. Please note that it should refer to current scientific reports and not just repeat the results with a mediocre reference to the existing literature.
- Please explain how missing data was handled and justify the choice of the CRP/Alb = 8 threshold (Youden vs. literature).
- Please increase the resolution of the figures.
- Expand the Limitations section to include the lack of a comorbidity index and the study's retrospective nature; propose a plan for future prospective studies.
- Standardize the bibliography by the Nutrients format and add the latest meta-analyses from 2024–2025 on malnutrition and inflammation in geriatric patients to the discussion.
- Please review the Similarity Report and paraphrase the text from the literature. The article still contains entire sentences taken directly from other articles. The similarity index is still far too high.
Best regards,
The reviewer.
Author Response
Comment 1. According to the latest guidelines from geriatric societies, the WHO (2021), and the style guides of leading publishers, the term “elderly” is now considered potentially discriminatory and reinforcing age stereotypes. Its use (especially in the title) can be interpreted as ageism and constitutes a serious barrier to publication in a prestigious scientific journal. It is strongly recommended to replace it with neutral, dignified terms: older adults, seniors, and older patients. Please consistently use these terms in the title, abstract, main text, and keywords to avoid unintended stigmatization and to meet the expectations of the Nutrients editorial board and the broader academic community
Response 1: Thank you very much for your observation. We replaced “elderly” with “older patients” throughout the title, abstract, keywords, and main text.
Comment 2. Expand the Discussion section. Please note that it should refer to current scientific reports and not just repeat the results with a mediocre reference to the existing literature.
Response 2: We sincerely thank you for your constructive feedback. The Discussion was expanded with recent cohort studies and meta-analyses (2024–2025) on nutrition state and inflammation in older patients, highlighting how our findings align with and extend current evidence. We have also divided the Discussion paragraph into subparagraphs to make it easier to read and understand.
Comment 3. Please explain how missing data was handled and justify the choice of the CRP/Alb = 8 threshold (Youden vs. literature).
Response 3: Patients with missing admission laboratory data were excluded from analysis (listwise exclusion). No imputation was applied. This has been specified in Materials and Methods, Section 2.1. Optimal cut-offs were determined using the Youden Index from ROC curve analysis. This clarification was added in Materials and Methods, Section 2.3.
Comment 4. Please increase the resolution of the figures.
Response 4: All figures were replaced with high-resolution versions to ensure clarity.
Comment 5. Expand the Limitations section to include the lack of a comorbidity index and the study's retrospective nature; propose a plan for future prospective studies.
Response 5: The Limitations section has been expanded to include: retrospective design, lack of comorbidity indices, absence of nutritional and functional assessments, and single-center design. We also outline plans for future prospective multicenter studies.
Comment 6. Standardize the bibliography by the Nutrients format and add the latest meta-analyses from 2024–2025 on malnutrition and inflammation in geriatric patients to the discussion.
Response 6: The entire reference list was standardized according to Nutrients guidelines.
Comment 7. Please review the Similarity Report and paraphrase the text from the literature. The article still contains entire sentences taken directly from other articles. The similarity index is still far too high.
Response 7: The manuscript, with the assistance of the Assistant Editor, underwent two careful revisions to reduce the duplication rate. We carefully revised and paraphrased all sections where overlap with published literature was identified. The revised manuscript now contains original phrasing throughout.
Round 2
Reviewer 1 Report
Comments and Suggestions for Authors
They have responded very well, and the manuscript has been improved now. I think this paper deserves to publishment.
Reviewer 3 Report
Comments and Suggestions for Authors
Dear Authors,
Thank you very much for the significant improvements you have made to your manuscript.
Best regards,
The reviewer.